# Development of an Application for Electronic Retrieval of Patient and Sample Information in Latin American Regions with a High Incidence of Gallbladder Cancer

**DOI:** 10.3390/jpm12091476

**Published:** 2022-09-09

**Authors:** Sergio Guinez-Molinos, Jaime Gonzalez Diaz, Carol Barahona Ponce, Justo Lorenzo Bermejo

**Affiliations:** 1School of Medicine, University of Talca, Talca 3460000, Chile; 2Statistical Genetics Research Group, Institute of Medical Biometry, University of Heidelberg, 69120 Heidelberg, Germany

**Keywords:** e-data collection, gallbladder cancer, personalized prevention, biorepository, collaborative research

## Abstract

The European–Latin American Consortium towards Eradication of Preventable Gallbladder Cancer, EULAT Eradicate GBC, is collecting high-quality data and samples in four Latin American countries with high gallbladder cancer incidence (Argentina, Bolivia, Chile, and Peru) to build a unique biorepository integrated into a tailored IT platform, to identify, validate, and functionally characterize new risk biomarkers, and to develop prediction models that integrate epidemiological and genetic–molecular risk factors. We decided to develop an application for electronic data collection to facilitate the retrieval of sociodemographic, clinical, lifestyle, dietary, and sample-related information from 15,000 Latin American study participants. The application EULAT eCollect will facilitate the work of study nurses, reduce time spent by participants, limit the use of paper and ink, minimize costs and errors associated with filling out written forms and subsequent digitisation, and support the monitoring of local recruitment rates and data quality. We describe in this article the design and implementation of the EULAT eCollect application, which started with the specification of functional and non-functional requirements, and ended with the implementation and validation of four separate application modules: Socio-Demographic Interview, Sample Information, Case Report Form, and Food-Frequency Questionnaire. We present both general and technical results, and our experience with the free and open-source software, Open Data Kit (ODK), which may be of interest for future related research projects, especially those on personalised cancer prevention carried out in low- and middle-income regions.

## 1. Introduction

Gallbladder cancer (GBC) is the most common neoplasms of the biliary tract [1,2]. GBC tumors are often diagnosed late, and GBC presents with the poorest prognosis of all gastrointestinal and hepatobiliary cancers [3]. The disease is particularly frequent in South America, where Bolivia, Chile, north-western Argentina, and southern Peru show high prevalence (Figure 1) [2]. Bolivia and Chile’s Araucanía region, the heartland of the Mapuche indigenous people, present the highest GBC mortality rates in the world [3,4,5,6]. Given the high prevalence and mortality due to GBC in South America, it is important to characterize the genetic, epidemiological, and biopsychosocial factors associated with GBC risk in these regions.

Among the risk factors associated with GBC, female gender, age, gallstones (cholelithiasis), genetic ancestry, obesity, multiparity, and socioeconomic and educational levels are the most important [1,4]. The limited ability to modify these risk factors and the complexity of individual risk profiles complicate GBC prevention, especially in Latin America [2]. Taking into account the high incidence of GBC in neighbouring South American countries (See Figure 1), there may be specific risk factors in the region that have not yet been identified [6]. Further research is urgently needed to identify novel GBC risk factors and better understand the epidemiology and molecular biology of the disease to improve the prevention and treatment [2].

Currently, there are no tests that reliably detect GBC early enough to be useful as screening tools [4,7]. Due to the high incidence of GBC, its adverse prognosis, and the difficulty in accurately predicting GBC risk, GBC is a major public health problem in high-incidence regions of South America. In this context, the identification of biomarkers to predict GBC would allow a personalized indication for preventive gallbladder removal (prophylactic cholecystectomy), thus improving the efficacy of this preventive measure. Because early GBC symptoms are inconspicuous and nonspecific, early diagnosis of GBC is difficult, and only around 20% of tumors are diagnosed at an early stage [7,8]. GBC tumors often spread to the liver and adjacent organs, and the overall median survival time for patients with advanced GBC is only 6 months. Given the high prevalence of cholelithiasis in South America, especially among women with a high proportion of Native American ancestry, the strong association between cholelithiasis and GBC, and the high incidence, late diagnosis, and poor prognosis of GBC despite the most aggressive surgery, there is general consensus that the most effective strategy for this cancer should be eminently preventive [4,6,8].

A European–Latin American research consortium towards eradication of preventable GBC–EULAT Eradicate GBC—has been established with funding from the European Union’s Horizon 2020 programme to identify factors related to the development of GBC, to find biomarkers in blood, saliva, urine, and faeces that allow individual risk prediction and early diagnosis of the disease, and to facilitate the development of better primary, secondary, and tertiary prevention strategies [9]. The consortium is collecting data and samples from GBC cases and controls in Argentina, Bolivia, Chile, and Peru, which will be combined with data from the largest European cohorts to ultimately contribute to the eradication of preventable GBC. The generated information will be essential for (a) identifying geographic, environmental, lifestyle, ethnic, and molecular differences in GBC risk and prognosis, and (b) translating the identified differences into applied GBC medicine, particularly the development of optimized GBC prevention programs tailored to the specific characteristics of different national health systems.

The stated goals of the EULAT Eradicate GBC project can only be achieved through collaboration between European and Latin American researchers, clinicians, governmental authorities, and representatives of cancer-patient societies. We plan to leverage samples and data from European cohorts and registries, and collect and analyze data and samples from 15,000 Latin American participants to build a unique European–Latin American GBC biorepository complemented by a customized IT platform applying a translational and multidisciplinary approach. It is challenging to collect high-quality socio-demographic, clinical, lifestyle, dietary, and sample-related information for this large number of study participants, mainly from low-income regions in four South American countries, which led us to develop a customized computer application to support not only the data-collection process but also the monitoring of local recruitment rates and data quality. The application for electronic data collection in the EULAT Eradicate GBC project (EULAT eCollect for short) was designed to facilitate the work of study nurses by, for example, displaying different questions depending on the characteristics of patients and samples, reduce the time invested by study participants in the interview, which is particularly important for patients in poor health, limit the use of paper and ink, and minimize costs and errors associated with the written completion of forms and subsequent digitisation by medical documentalists. Important requirements for EULAT eCollect were the ability to collect and store data without a stable internet connection, and the secure transfer of data to the storage server when the internet is available.

This article describes the process of designing and developing the application EULAT eCollect to facilitate the collection, storage and future analysis of high-quality information from patients and their families. EULAT eCollect consists of four separate modules: Socio-Demographic Interview, Sample Information, Case Report Form, and Food-Frequency Questionnaire. We first present the specification of the functional and non-functional requirements, then the design of the application architecture, and conclude the manuscript with a description of the four implemented modules and the usefulness of EULAT eCollect to monitor recruitment rates and data quality.

## 2. Materials and Methods

The process of designing and developing an information technology (IT) infrastructure for electronic data collection was based on typical software engineering methodology [10]. The definition of requirements was the first step in the development of the application EULAT eCollect. This step was fundamental because it laid the foundation for all subsequent work and had a direct impact on the success of the application under development [11]. The primary source for the definition of requirements were the documents of the research project EULAT Eradicate GBC [9], from the grant proposal to the standardized operating procedures. Detailed information on ethics approvals can be found in the section Institutional Review Board Statements.

EULAT eCollect was designed to support patient-data collection by incorporating the functional and non-functional requirements needed to make it easy and secure for the recruiting personnel. Study participants are interviewed by trained nurses before surgical gallbladder removal at the participating hospitals, or in the participant’s home (some patients prefer this alternative). After the complete process of informed consent, which includes a detailed description of study objectives and participant rights, clarification of any questions or concerns the potential participant may have, and signing of the consent form, recruitment personnel apply a socio-demographic questionnaire. Interviews are always conducted in a private room to protect data confidentiality.

The specification of functional requirements (see Table 1) included the definition of tasks to be performed using the web application, such as the application of socio-demographic interviews adapted to language particularities in Argentina, Bolivia, Chile, and Peru, the retrieval of information on collected samples (blood derivatives, urine, faeces, and saliva), the completion of patient case reports with clinical information, and the application of food-frequency questionnaires. The engineering process involved elicitation, documentation, and analysis [10], which were primarily based on the grant application for the EULAT Eradicate GBC project [9]. The process was complemented by structured discussions about the functionalities and attributes necessary to build a web application with the project coordinating team.

The non-functional requirements refer to technical aspects of the web application, such as usability, information security, access to and quality of the internet, and the hardware performance [10]. Regarding the non-functional requirements, usability is essential for designing interfaces and improving user interaction with the application to provide a satisfactory experience [12]. The design of the usability considered the project’s specific characteristics and technological aspects, and the staff in charge of data collection [13]. In addition, information security and patient-data integrity are fundamental non-functional requirements. For example, the developed application complies with the General Data Protection Regulation (GDPR) of the European Union [14] and the German Federal Office for Information Security [15].

The project contemplates collecting, storing, and analysing sensitive information to reach the project’s objectives, which implies a significant challenge of keeping confidentiality and protecting participants’ data. The information collected to improve the prevention and early diagnosis of GBC in Latin America contains sensitive data. Part of the data collected is considered clinical information, and the legislation of the participating countries mandates that this type of information is stored, processed, and used appropriately [16]. Accordingly, we implemented the application EULAT eCollect using the modular, open-source toolkit Open Data Kit (ODK), which enables organisations to create applications for use in resource-constrained environments and allows data capture using Android devices such as tablets and mobile phones, or a web browser [17]. ODK is one of the leading data-collection solutions available and offers two main tools: ODK Collect and ODK Central [17,18]. ODK Collect is an open-source Android application that replaces paper forms. It supports many questions and answer types, is designed to work well without network connectivity and facilitates the uses of different device sensors to retrieve a wide range of information, e.g., the tablet’s camera or a Bluetooth-connected scanner can be used to read sample barcodes and images (pictures/PDFs from blood analyses and pathology reports). The server ODK Central manages user accounts and permissions, stores form definitions, and allows data-collection clients to download and upload forms.

### Design of the Four Application Modules

The application EULAT eCollect for electronic retrieval of patient and sample information was developed and is currently being applied to guarantee standardized data collection at all project recruitment sites. The following complementary activities were also carried out to ensure the establishment of a harmonized project biorepository: development and application of technical operational guidelines, training of recruitment staff in biobanking and biorepository procedures, establishment of a data and sample quality assurance programme, and development and establishment of automated quality control and standardized data-management procedures.

EULAT eCollect was organised into four separate modules, taking into account that the ultimate objective of the EULAT Eradicate GBC project is to identify major modifiable dietary and lifestyle risk factors (smoking, alcohol, and physical activity), body size/shape measurements, and reproductive and hormonal factors, and to evaluate possible differences by gender and ethnicity. In a first step, the association between dietary and lifestyle characteristics and GBC risk will be assessed using multiple conditional logistic regression models. Biologically plausible effect-modifiers will then be considered (specific nutritional factors, metabolic alterations, and circulating levels of inflammatory markers; where biomarker data are already available or being generated) by investigating potential interactions between biomarkers and dietary/lifestyle risk factors. With these goals in mind, the following application modules were considered:

**Module 1: Socio-Demographic Interview.** The development of this module combined our own experience from previous GBC research [19] with the design of the latest Chilean National Health Survey [20].

**Module 2: Sample Information.** The aim of this module was to facilitate the collection of relevant information on different types of samples (blood, urine, faeces, and saliva). The type of samples collected depends on the type of study participant, and our application simplifies the work of staff processing study samples by asking for participant-specific information. The data collected includes transport, processing, and storage conditions, which greatly facilitates the automatic generation of pre-analytical sample quality indicators.

**Module 3: Case Report Form.** This module was developed to collect clinical patient information, the characteristics of resected gallbladder specimens, and treatment information. We tried to incorporate the research interests of all collaborating clinicians, and also took into account the latest guidelines for pathology reporting [21].

**Module 4: Food-Frequency Questionnaire.** This module was based on a validation study of a self-administered food-frequency questionnaire for Argentina, Chile, and Uruguay [22].

The four modules were implemented using the free and open-source software ODK [17]. The widespread use of this software in low-income regions with irregular and poor internet access, the advantages associated with free and open-source software, and the privacy and security of personal data provided by ODK motivated our decision to use this software [18]. In the ODK environment, a client (ODK Collect) can be installed on Android devices to share and fill user-defined forms (Modules 1–4 in EULAT eCollect), which are sent to a secure server when an internet connection is available (Figure 2). The Java web application ODK Aggregate then permits the submitted forms to be stored as a PostgreSQL database.

After creating an application prototype, EULAT eCollect modules were validated with the help of an R script developed for this purpose, which was used to calculate quantitative and qualitative indicators for the data collected. In June 2021, we analyzed a pilot dataset using both the R script and manually calculated indicators. The few discrepancies (e.g., typos in field names and exemplary data entries, or aliquot volume categories) were identified and corrected, leading to the final version of EULAT eCollect, which is now fully operational and has been used for almost a year to recruit gallstone and GBC patients in Argentina, Bolivia, Chile, and Peru, with a positive experience so far. A bioinformatics engineer (J.G.D.) was responsible for the technical deployment with the active support of the head of the Biomedical Informatics Laboratory at the Universidad de Talca (S.G.M.). The project coordinator (J.L.B.) and the project manager in Latin America (C.B.P.) were in charge of formulating and verifying the application requirements, with excellent support from the Biobank of the Universidad de Chile during the validation phase. The entire team spent six months creating the application prototype and another two months making the necessary modifications during the validation phase.

## 3. Results

### 3.1. Functional and Non-Functional Application Requirements

A fundamental input for the design of EULAT eCollect was the requirements identified and listed in Table 1. Each application module had specific functionality requirements. In addition, non-functional requirements for usability, regularity of internet access, and data security were also considered. Taking these requirements into account, the interfaces of the individual modules, their main areas, and the way they were to be filled were designed and implemented.

### 3.2. Data Collection and Upload to the Application Server

After defining the functional and non-functional requirements for EULAT eCollect, we started its design and implementation. The application was developed under a model–view–controller architecture to separate the views of the data model and the business logic [23]. Figure 2 shows the general architecture of EULAT eCollect and the software used. Within the EULAT Eradicate GBC project, data can be electronically collected using mobile devices or a web browser. Mobile devices include tablets and mobile phones, which belong to the basic equipment of the recruitment staff and allow them to install the application ODK Collect, and then download, fill in, and send the four forms developed for the project [17]. ODK Collect allows data capture through android devices and works without internet connectivity, allowing data collection in remote locations. With ODK Collect, various sensors on the device can be used to retrieve a wide range of information. For example, the tablet camera or a Bluetooth-connected scanner can be used to capture sample barcodes and various images, such us pictures of blood tests and pathology report PDFs.

EULAT Eradicate GBC data can also be collected via a web browser on a desktop or mobile device (e.g., iOS device, see Figure 2). As ODK Collect is primarily designed to provide forms to data collectors using the ODK Collect application, an open-source plug-in (Enketo) is also used to accept submissions from other platforms and devices [24]. These applications are logically partitioned so they can be used for lightweight mobile hybrid applications and web applications. The Enketo plug-in can be accessed from any web browser, in both online and offline modes. Once the device has an internet connection, the collected data are sent to our ODK Central server, where the data are safety stored.

### 3.3. Data Security and Protection

Different regulatory frameworks apply to the storage and use of information collected within EULAT Eradicate GBC in Argentina, Bolivia, Chile, and Peru. However, the laws in force in the four countries focus on the treatment of information stored in clinical records. The information collected with the aim of improving the prevention and early diagnosis of GBC in Latin America contains sensitive data, part of which is clinical data that must be properly stored, processed, and used according to national regulations. In addition to the developed application for electronic data collection EULAT eCollect, an IT platform that stores and provides authorised access to the collected data is expected to complement our unique biorepository of GBC samples from Europe and Latin America as a primary resource for translational research in individualised prevention, personalised early detection, and precision therapy of GBC. For these reasons, data security and protection are critical non-functional requirements for EULAT eCollect, which were fulfilled as follows:

Hardware security was provided at two different levels: physical security and personal usernames and passwords. The first level, physical security, refers to restricting or rejecting access to the devices and computers used to collect and store the data; only authorized personnel can handle this equipment. Servers and networks are housed in a suitable building with controlled access. Data-collection devices are kept in restricted hospital areas where only project nurses and staff can access them. At the second level, study nurses must enter their personal username and password to collect data electronically using tablets and desktops, and to transfer these data to the server. Furthermore, the ODK Collect application requires additional username/password authentication for each individual data collector in the recruitment centres. On the server side, access to computers, databases and platforms is protected by different personal usernames and passwords. 

Data exchange takes place via a secure encrypted data exchange protocol HTTPS. The use of SSL certificates prevents unwanted observation during data exchange from the collection devices to the central ODK server. In addition, the ODK tools use digest authentication to secure username/password authentication, with an encrypted form of the password stored on the server. During the authorization process, the encrypted form is sent over the internet instead of the raw password. To send surveys from Enketo or ODK Collect to the ODK Central server, username/password authentication is required. Anonymous user submissions are not allowed.

### 3.4. Features of the Module 1: Socio-Demographic Interview

The first module of EULAT eCollect facilitates the collection of sociodemographic information from study participants, taking into account the specificities of Argentina, Bolivia, Chile, and Peru, such us language use, health-care system, salary currency, and socioeconomic level. A substantial effort was made to capture all major established and potential GBC risk factors, using high-level measurement scales whenever possible (e.g., interval and ratio scales were preferred over ordinal scales) and to guarantee consistency with the Chilean National Health Surveys. Standard answers were formulated to avoid typing errors and mandatory fields were defined to limit the amount of missing information.

After the informed consent, recruitment personnel enter their authentication credentials and select the type of participant. The application EULAT eCollect displays specific questions for four types of participants: (1) cholelithiasis patient, (2) GBC patient recruited before any treatment (surgery, chemotherapy, or radiation therapy), (3) GBC patient recruited after treatment, and (4) first-degree relative (father, mother, brother, sister, son, or daughter) of a GBC patient (proband) in a family with multiple members affected by GBC.

The patient interview is structured into eight to nine sections: (I) basic information, e.g., participant code, recruiter identifier, and type of participant, (II) demographic information, (III) socioeconomic information, (IV) lifestyle factors, (V) individual health history, (VI) family health history, (VII) body measurements, (VIII) physical activity, and (IX) gynaecological information (for women only). In Bolivia and Peru, an additional item on coca-leaf use was included in the interview. Visual aids are used when conducting the interview, especially for questions on lifestyle and physical activity (Figure 3), to standardise responses, such as portions of food and drink, and intensity of physical activity. The EULAT eCollect application significantly reduces the duration of the interview to about 30 min, compared to around 45 min for written interviews, which is an important advantage especially for frail or elderly patients.

### 3.5. Features of Module 2: Sample Information

After biospecimen collection, samples are processed, aliquoted, and stored in compliance with all relevant ethical and legal regulations for biomedical research, and to the standardized operating procedures of the EULAT Eradicate GBC research project. The second module of the EULAT eCollect application facilitates the electronic retrieval of sample information from the donation of blood, urine, faeces, and saliva, through transport to the laboratory, to sample processing and storage, enabling the automatic creation of customized sample inventories for shipment, receipt, and further processing, including pre-analytical quality indicators for the different types of stored samples (whole blood, buffy coat, serum, plasma, urine, faeces, saliva, and DNA). Recruitment staff collect different biospecimens depending on the type of participant. For example, only saliva is collected from GBC patients who have been cholecystectomised or started chemotherapy. In contrast, cholelithiasis patients donate blood, urine, and faeces before cholecystectomy. After authentication, the staff responsible for sample processing and storage, which may be different from the staff who collected the samples, selects the type of sample in EULAT eCollect, which then displays specific questions depending on the type of sample and previous answers (Figure 4). For example, if the user selects “Deviations from standardized operating procedures = Yes”, the application asks for the type of deviation. The collected information includes, for example, date and time of sample collection, sample characteristics (e.g., volume, colour, and general appearance), protocol deviations, and characteristics of stored aliquots (e.g., time of freezing at −80 °C and position in storage box). A strength of EULAT eCollect is the ability to scan barcodes of boxes and tubes used to store aliquots with the camera of the device used (e.g., tablet or smartphone) or a barcode reader. This feature minimises barcode errors and considerably reduces the amount of time required for sample-processing staff to document sample characteristics.

### 3.6. Features of Module 3: Case Report Form

This module of the application facilitates the collection of clinical patient and anatomical–pathological gallbladder information, taking into account the research interests of participating clinical partners (pathologists, surgeons, oncologists, and gastroenterologists), particularly in view of future studies on early diagnosis, predictive and prognostic biomarkers, and personalised treatment. To illustrate, information collected on GBC patients includes method of diagnosis, tumor stage, lymph node involvement, presence and location of metastases, histological type of the tumor, and comorbidities, as well as information on surgical procedures (e.g., date and outcomes) and treatment (e.g., type of chemotherapy or radiotherapy). Central to the design and implementation of this module were the latest guidelines for pathology reporting [21]. The case report form is structured into six sections: (a) General information, e.g., date of data entry, (b) Patient information, e.g., concomitant diseases, (c) Primary clinical information, e.g., type of diagnostic imaging, (d) Surgical information, e.g., type of cholecystectomy, (e) Pathological report, and (f) Gallbladder information, e.g., length, width, and appearance of the hepatic and serosa surfaces. A convenient feature of this module of EULAT eCollect is the ability to upload images and PDFs of relevant documents, for example blood tests and the complete pathology reports.

### 3.7. Features of Module 4: Food-Frequency Questionnaire

The fourth module of EULAT eCollect enables the electronic application of a food-frequency questionnaire (Figure 5) validated for Argentina, Chile, and Uruguay by Natalia Elorriaga at the South American Center for Cardiovascular Health and colleagues at Universidad de la República in Montevideo, Uruguay and Universidad de la Frontera in Temuco, Chile, which is one of the recruitment sites of the EULAT Eradicate GBC project [22]. The questionnaire has the original structure of the Spanish version of the Dietary History Questionnaire I developed by the US National Cancer Institute, with a list of food and beverages adapted to those commonly consumed in Argentina, Chile, and Uruguay, with food names, examples of brand names, and portion sizes adapted to the usual denominations in each country, and some questions reworded for persons with a low level of education (primary school or less) [25]. In addition to 126 separate food items with portion sizes in units and household measures such as cups or spoons, seven questions on the use of low-fat and low-sugar foods, four summary questions, and nine questions on food supplements, individual intakes of energy, twenty nutrients, and fruit and vegetable consumption can be answered using e.g., the LATINFOODS database (latinfoods.inta.cl) to assess possible associations with GBC risk and prognosis, possibly in combination with metabolic alterations and circulating levels of inflammatory markers.

### 3.8. Monitoring Patient Recruitment in the EULAT Eradicate GBC Project with EULAT eCollect

An important feature of EULAT eCollect is its usefulness for real-time tracking of patient recruitment in each recruitment site of the EULAT Eradicate GBC project using qualitative and quantitative indicators. Briefly, the forms sent by recruitment personnel to the project’s secure server are processed using R scripts to generate 19 indicators on the number of patients recruited, and on the quality of collected samples and socio-demographic, clinical, and dietary information.

Quantitative indicators include the cumulative number of forms collected; stratified by form type (e.g., cumulative number of socio-demographic interviews); the number of patients with all forms completed, as pathology reports required to complete case report forms are usually available 3–4 weeks after cholecystectomy; the distribution of participants according to participant type at interview (cholelithiasis, prospective, or retrospective GBC patient, or member of a family with several GBC cases), the total number of recruited participants per month; the distribution of participants according to type/s of donated samples (e.g., blood, urine, and faeces, or saliva only); and the distribution of patients according to anatomy reports (e.g., patients with cholelithiasis who showed GBC after inspection of the resected gallbladder). In addition to reports in tabular format (e.g., in Excel), R scripts also generate useful figures that facilitate comparison of the recruitment progress between recruitment sites (Figure 6).

Qualitative indicators include the proportion of patients with all forms completed, the proportion of incomplete, non-compulsory fields in the four modules of EULAT eCollect (overall per module, and also per module section, the average number of specific types of aliquots per participant (e.g., number of serum aliquots), and the proportion of samples showing deviations from the sample processing protocol. Thresholds were defined and validated for specific indicators (e.g., minimum average number of serum aliquots per participant) and indicators are highlighted in the monitoring reports when thresholds are exceeded.

## 4. Discussion

Research on GBC has been largely neglected, as this aggressive disease is relatively rare in most high-income countries. Several South American regions including northeaster Argentina, Bolivia, Chile, and southern Peru, have a high GBC incidence [3,4,5]. Inhabitants of these regions could be exposed to particular genetic, environmental, and lifestyle risk factors, and large-scale population-based studies that investigate differences in risk exposure between low (Europe) and high (South America) incidence regions are needed to identify GBC risk factors and provide information for personalized primary, secondary, and tertiary GBC prevention.

Defining functional and non-functional requirements was the first step in designing the application EULAT eCollect. Functional requirements included minimizing interview time, especially for frail study participants, avoiding errors when creating the sample inventory using barcode readers or mobile phone cameras, taking into account the latest pathology guidelines and applying previously validated food-frequency questionnaires. Non-functional requirements comprised the ability to collect data without internet connection and the secure transfer of data, which takes place via HTTPS protocols. Digest authentication with an encrypted form of the password stored on the server is used to secure username and password authentication.

EULAT eCollect enables electronic data collection for the EULAT Eradicate GBC project via mobile devices (e.g., tablets and mobile phones). Once the application is installed, staff recruiting study participants and processing samples can download, fill in, and send the four forms developed for the project. The first module of EULAT eCollect facilitates the recording of the main established and potential GBC risk factors in accordance with the Chilean National Health Surveys. The second module of the application facilitates the electronic retrieval of sample information from biospecimen donation to sample storage, enabling the automatic generation of sample inventories that include pre-analytical quality indicators. A practical feature of the third module of EULAT eCollect is the ability to upload images and PDF files of blood tests and pathology reports for case report forms. Finally, the Food-Frequency Questionnaire embedded in EULAT eCollect allows for efficient collection of comprehensive nutritional information. An important activity in the EULAT Eradicate GBC project is the monitoring of patient recruitment, which is greatly facilitated by the automatic generation of quantitative (e.g., the number of patients with completed forms) and qualitative indicators (e.g., the average number of aliquots per participant) through the EULAT eCollect application.

The free and open-source ODK software was chosen for the implementation of EULAT eCollect. ODK provides convenient tools for developing a data-collection system with an integrated database [17]. The flexibility that ODK offers for data storage and the advantages of open source software led us to use this software. We also considered the possibility of using alternative platforms such us Research Electronic Data Capture (REDCap) [26]. REDCap is a web application used in many research projects that allows investigators to easily create and manage research databases. Table 2 compares various features of ODK and REDCap. One advantage of ODK for the EULAT Eradicate GBC project was the ability to use mobile devices for electronic data collection in remote locations where internet access is deficient or non-existent—internet bandwidth fluctuates widely between participating recruitment sites. Another important feature of ODK for the project was the use of sensors (e.g., camera) on mobile devices to read barcodes and facilitate an efficient sample processing and storage. ODK also enables the programming of data checks in the data-collection instruments, which is essential to safeguard the quality of data recorded in different centres by multiple professionals.

The collection, storage and analysis of sensitive information to reach the objectives of the EULAT Eradicate GBC project is a major challenge—confidentiality must be maintained, participants’ data must be protected, and the participating countries have different legislation for the use of sensitive patient data. Therefore, we have paid special attention to information security in the development of EULAT eCollect. The four application modules were built in such a way that the identity of the patients remains anonymous—components of data records that would allow participant identification are replaced by pseudonyms on-site, and only treating physicians know patient identity. In addition, data are transferred via HTTPS protocols for which a key and certificate are generated based on regulatory requirements, firewalls restrict access to the servers where data are stored, and administrative access is limited to the management of the institution’s intranet.

## 5. Conclusions

The EU-funded project EULAT Eradicate GBC is collecting high-quality data and samples from 15,000 study participants to identify geographic, environmental, lifestyle, ethnic, and molecular differences in GBC risk and prognosis, and to translate the results into optimised GBC-prevention programmes considering the specificities of national health systems. The collection of high-quality information for this large number of participants in low- and middle-income regions is challenging and motivated the design and development of a customized computer application, EULAT eCollect, which we describe in this article.

EULAT eCollect can be easily adapted or used directly for electronic data collection in similarly oriented research projects, and we explicitly offer our support in this direction. For example, current and future research projects on the prevention of gastric, colorectal, and biliary malignancies in Latin America could benefit from our experience in designing and implementing the application. Modules 1, 2, and 3 will probably require some modification to take into account established and potential risk factors for the disease under investigation, the type of samples and aliquots collected in the project, and clinical characteristics according to generally accepted guidelines, but Module 4 could be used directly after clarification of data storage. We also offer to share, interested researchers, our list of quantitative and qualitative indicators for monitoring of the participant recruitment.

## Figures and Tables

**Figure 1 jpm-12-01476-f001:**
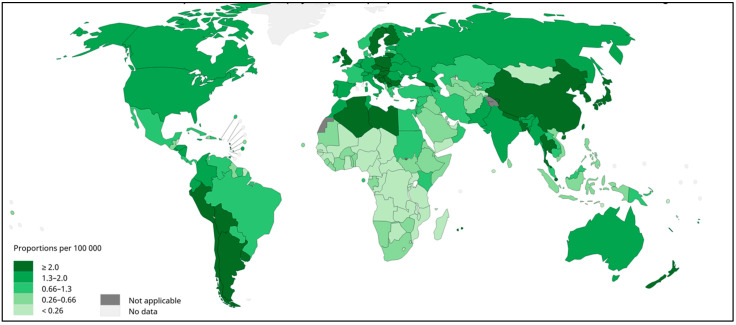
Gallbladder cancer incidence in both sexes, worldwide, 2020. Estimated age-standardized rates (World) per 100,000 per year. Data source: GLOBOCAN 2020. WHO, IARC: https://gco.iarc.fr/today/ (accessed on 23 July 2022).

**Figure 2 jpm-12-01476-f002:**
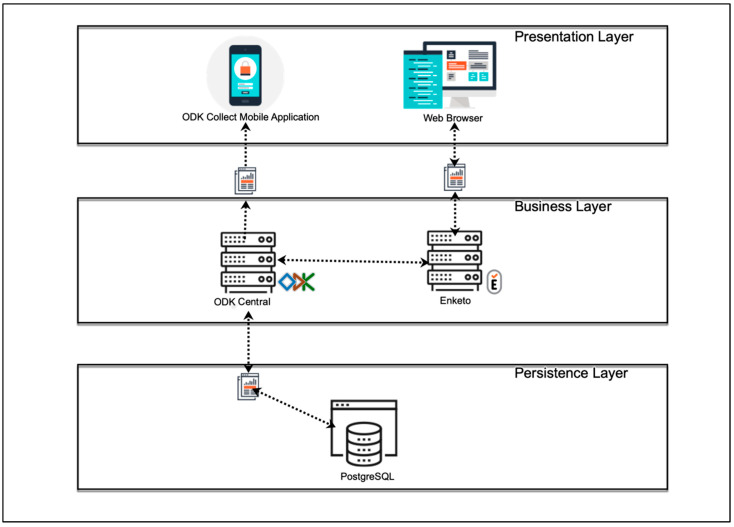
Software architecture of the developed application EULAT eCollect for electronic retrieval of patient and sample data in Latin American regions with a high incidence of GBC (Bolivia, Chile, north-western Argentina, and southern Peru).

**Figure 3 jpm-12-01476-f003:**
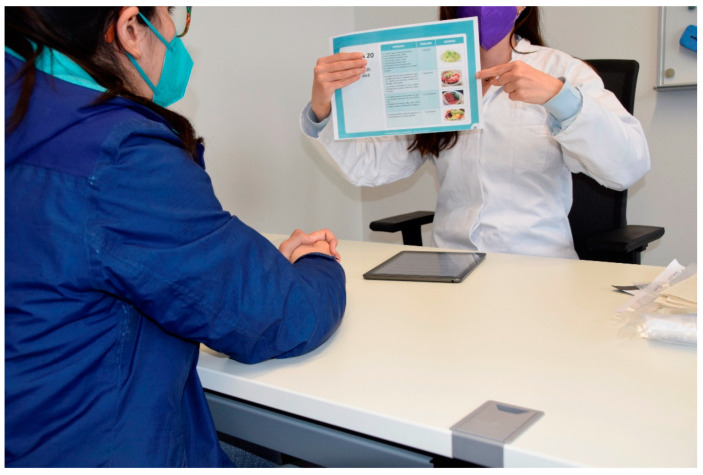
Visual aids are used during the interview with the first module of the EULAT eCollect application (Socio-Demographic Interview), which displays different questions depending on the type of participant, patient characteristics (e.g., gender), and previous answers, allowing for a significant reduction in interview time.

**Figure 4 jpm-12-01476-f004:**
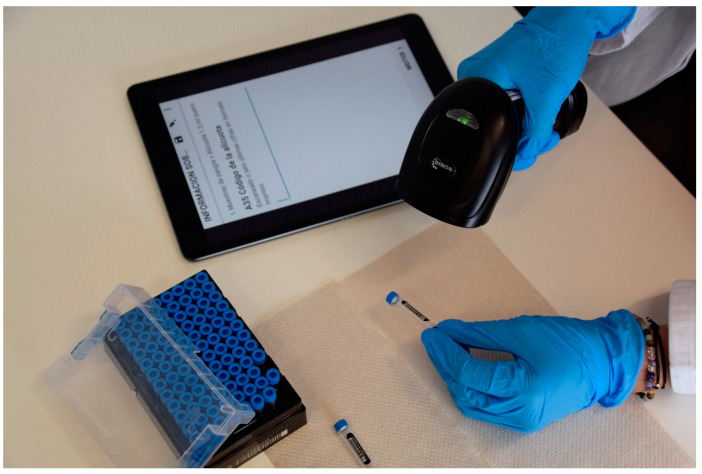
Second module of EULAT eCollect (Sample Information). The application allows barcodes to be scanned using a barcode reader or the camera of the device on which EULAT eCollect is installed (e.g., tablet or smartphone), minimising errors and reducing the time spent documenting sample characteristics during sample processing.

**Figure 5 jpm-12-01476-f005:**
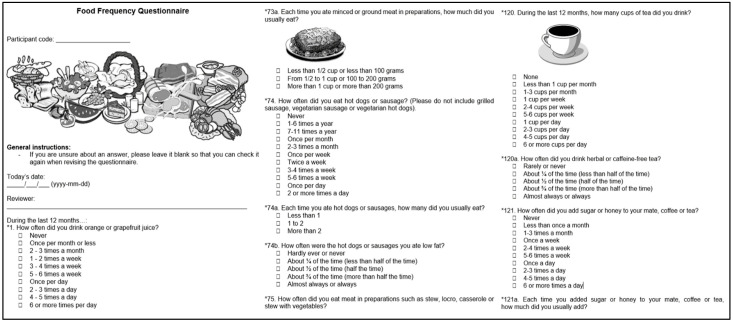
English translation of exemplary screenshots of the food-frequency questionnaire integrated in EULAT eCollect.

**Figure 6 jpm-12-01476-f006:**
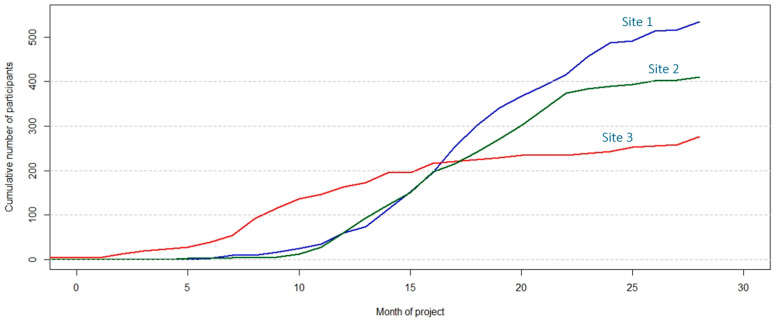
Graphical representation of the cumulative number of participants recruited in three exemplary recruitment sites to monitor the progress of the EULAT Eradicate GBC project against planned numbers.

**Table 1 jpm-12-01476-t001:** Summary of the application requirements as outcome of the inspection of EULAT Eradicate GBC documents, complemented by extensive discussions of the whole development team.

Module/Type of Requirement	Detailed Requirements
1. Socio-Demographic Interview/Functional	- Capture established and potential GBC risk factors, where possible, using high-level standardized scales (for example, ratio rather than ordinal scale) - Consistency with the Chilean National Health Survey - Minimize interview time, especially for frail and elderly study participants - Easy generation of basic reports for project monitoring, e.g., number of patients recruited per month and interview completeness
2. Sample Information/Functional	- Complete coverage of sample information, from sample donation to transport to the lab, processing and storage characteristics, including pre-analytical quality indicators, for different sample types (blood, urine, faeces, and saliva) - Minimize time and errors in sample inventory using a barcode reader or mobile device camera - Easy generation of comprehensive sample inventories for sample shipment, receipt and further processing, especially with regard to sample quality
3. Case Report Form/Functional	- Consideration of the latest pathology reporting guidelines - Capture patient, gallbladder, and tumor information, taking into account the research interests of clinical partners and future studies on prognosis, survival, and targeted therapy
4. Food-Frequency Questionnaire/Functional	- Application of questionnaires previously validated for the participating countries
All Modules/Non-functional	- Minimize the use of paper and ink, and costs and errors associated with double typing (First in writing, then digital documentation) - Securely collect and store data in the device without an internet connection, and securely transfer the data to a secure server when the internet is available. - Data access security (authorization, authentication, and data privacy)

**Table 2 jpm-12-01476-t002:** Key features of the free and open-source Open Data Kit (ODK) software used to implement the four modules of EULAT eCollect compared to the Research Electronic Data Capture (REDCap) web application, one of the leading platforms for capturing and processing clinical information.

	ODK	REDCap
Focus	Data-collection system with a database system added on	Database system with a data-collection system added on
Website	https://opendatakit.org (accessed on 30 June 2021)	https://projectredcap.org (accessed on 30 June 2021)
License	Free and open-source	Free to non-profit organisations and not open-source
Users	No restrictions	Institutional partner of the project
Low-Resource Mobile Data Collection	Yes	No
Data Checks Coding Into Data-Collection Instruments	Yes	No
Architecture	Suite of open-source tools for collection and data management	Web application to build online surveys and databases
Security	Authentication: DigestAuth Transport Layer: SSL/HTTPS	Authentication: LDAP/Shibboleth Transport Layer: SSL/HTTPS
Mobile App OS	Android (Web other OS)	Android, iOS
HIPAA-Compliant	No	Yes
Persistence Database	Relational (PostgreSQL, MySQL)	Relational (MySQL)
Design Forms	XLSForm standard	REDCap Data dictionary (CSV)
Offline Data Collection	ODK Collector and web version (Enketo)	REDCap Mobile App is required
Data Collection From Device Sensors	Yes	No

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
