# Peer review of "Development of an Application for Electronic Retrieval of Patient and Sample Information in Latin American Regions with a High Incidence of Gallbladder Cancer"

_jpm, 2022, doi:10.3390/jpm12091476_

Round 1

Reviewer 1 Report

The paper is well written and the topic would of real interest to the journal readers. The materials and methods are adequate, the research is properly done, and the conclusions are entirely sustained by the results.

Only few minor comments:

In line 42, the references should be in the format [3–6] not [4] [3,5,6].

In line 118, please replace applciation with application.

Reference no 13 appears not to be cited in the text. References should be numbered in there order of appearance in the text.

Figure 4, Figure 5 appears not to be cited in the text.

Reviewer 2 Report

Thank you for allowing me to peer review your manuscript. The research is interesting and once greater transparency and attention is paid to the methods and results section (especially regarding ethics and privacy and security) will prove invaluable to to readers.

The methods and results section require writing  clearly and succintly

There is no conclusion section included on the manuscript provided

I have made substantial comments on the document
